# Clinical Significance of Neutralizing Antibodies in COVID-19: Implications for Disease Prognosis

**DOI:** 10.3390/life15030429

**Published:** 2025-03-08

**Authors:** Sudem Mahmutoğlu Çolak, Tuba İlgar, İlkay Bahçeci, Esra Özkaya, Merve Hüner Yiğit, Hilal Durmuş, Feyza Atiş, Ayşe Ertürk, Zihni Acar Yazıcı

**Affiliations:** 1Departments of Infectious Diseases and Clinical Microbiology, Faculty of Medicine, Recep Tayyip Erdogan University, 53100 Rize, Türkiye; sdmmahmutoglu@gmail.com (S.M.Ç.); tuba.ilgar@erdogan.edu.tr (T.İ.); ayse.erturk@erdogan.edu.tr (A.E.); 2Departments of Medical Microbiology, Faculty of Medicine, Recep Tayyip Erdogan University, 53100 Rize, Türkiye; ilkay.bahceci@erdogan.edu.tr (İ.B.); hilal.durmus.96@hotmail.com (H.D.); feyza_atis20@erdogan.edu.tr (F.A.); 3Department of Microbiology, Faculty of Medicine, Karadeniz Technical University, 61080 Trabzon, Türkiye; esraozkaya@ktu.edu.tr; 4Departments of Medical Biochemistry, Faculty of Medicine, Recep Tayyip Erdogan University, 53100 Rize, Türkiye; merve.huner@erdogan.edu.tr

**Keywords:** neutralizing antibodies, fusion inhibition, COVID-19 severity, pathogenesis, mortality, vaccination

## Abstract

The pathogenesis of COVID-19 highlights a complex relationship between disease severity and neutralizing antibodies (NAbs). We aimed to investigate the relationships among the total NAb (tNAb) levels, the presence of potential neutralization antibodies (pNAbs), and the clinical outcomes of COVID-19 patients. Patients aged ≥18 years diagnosed between October 2021 and September 2022 were grouped by symptom severity. Blood samples were taken at two time points and data on demographics, epidemiology, and vaccination were recorded. The tNAbs and pNAbs were measured by an enhanced chemiluminescence assay and a surrogate virus neutralization test, respectively. The tNAbs of 68 and the pNAbs of 52 patients were analyzed. Twenty-six (38.2%) patients had severe infection. The 28-day mortality rate was 16.2% (n = 11). The tNAb levels in the control blood samples (CBSs) were significantly higher than those of the admission blood samples (ABSs) (*p* < 0.001). The statistical analysis showed no relationship between disease severity and pNAbs. Vaccinated patients had significantly higher tNAbs in the ABSs and CBSs (*p* < 0.001 and *p* < 0.001, respectively). The presence of pNAbs in the ABSs was correlated with a lower 28-day mortality (*p* = 0.026) and a milder disease course (*p* = 0.041). Although these findings support a correlation between tNAbs and disease severity and mortality, their presence seems to be unrelated to favorable clinical outcomes.

## 1. Introduction

The COVID-19 pandemic emerged in December 2019. Despite the intensive research since then on various aspects of the disease, there are still major gaps in our understanding of its pathogenesis.

The initial stage of viral infection occurs when the receptor binding domain (RBD) of the spike (S) protein of the virus binds to the angiotensin-converting enzyme 2 (ACE2) receptor on the host cell. Following entry, the cellular pathogen receptors recognize the virus, and the immune system is activated. The antiviral response involves the activation of IFN I and III and the downstream signaling pathways. It has been shown that in severely ill patients, the antiviral signaling pathways are interfered with by viral structural and non-structural products. Post intracellular replication, the virus is shed, and an antibody response ensues [1,2].

The RBD is the main site of action of neutralizing antibodies (NAbs) [3,4]. The affinity and titers of the NAbs determine their efficacy [5,6,7,8]. However, in the context of COVID-19, elevated serum NAb titers are not correlated with protective immunity in many cases [9,10,11]. Although numerous studies have demonstrated the efficacy of NAbs in viral neutralization assays, they fail to explain the serious illness and/or fatality in their presence at elevated titers [11,12]. In addition, the virus can evade NAbs by mutations. Despite this, neutralization activity is not completely lost in previously infected, vaccinated, or hybrid immune individuals, since the virus has to retain the receptor-binding motif within the RBD [1,13].

Several studies have demonstrated that the disease progresses more severely in patients with low NAbs and is milder in those with high NAbs [14,15]. It has also been observed that humoral immunity is more robust and NAb levels are elevated in individuals exhibiting severe disease symptoms [16]. In other words, disease severity determines the type and extent of the antibody response [17]. It has been postulated that the neutralization efficacy of anti-RBD NAbs is heterogeneous, with some being ineffective, hence the discrepancies on the role of high-titer NAbs in the protection and recovery from the disease [3,5]. The heterogeneity here refers to antibodies that bind to the RBD of the virus (tNAbs) but fail to hinder the docking of the receptor-binding motif of the virus to ACE2 (fusion inhibition). The antibodies binding to the receptor-binding motif of the virus are referred to as potential neutralization antibodies (pNAbs).

The anti-SARS-CoV-2 vaccines were developed and applied after the pandemic significantly affected the most vulnerable people in the population. Even after this, the efficacy of the vaccines provided conflicting outcomes. It has been shown that the antibody response generated through infection or after vaccination is inversely correlated to the severity of the disease [18]. Moreover, the vaccine effectiveness is better against symptomatic infection than asymptomatic infection [19]. However, it has also been demonstrated that the NAbs formed after vaccination are not a marker for protection from the disease [20]. The vast majority of people who fail to become ill post-viral exposure and those who become mildly ill lack NAbs [14,15].

At the conception of this project, the role of NAbs of all immunoglobulin classes (tNAbs) in the disease pathogenesis and/or protection was unclear. Therefore, we set out to investigate the correlation, if any, of tNAbs with the clinical course of the disease, as well as other hematological and biochemical parameters. The potential of the tNAbs for inhibiting the binding of the RBD to ACE2 receptor (pNAbs) was also investigated. The ‘recovery stage’ tNAb levels were significantly higher than those of the admission stage. Even though pNAb positivity was higher at the admission stage, the difference was not significant. No significant correlation was found between the clinical course of the disease and the tNAb or pNAb levels.

## 2. Materials and Methods

### 2.1. Study Population

Patients aged ≥18 years admitted to the pandemic wards of the Recep Tayyip Erdoğan University Research Hospital between October 2021 and September 2022 were included in the study. A COVID-19 diagnosis was confirmed by SARS-CoV-2 real-time polymerase chain reaction (PCR). The clinical group allocation of patients was as follows: Patients without symptoms were defined as asymptomatic, those with mild symptoms but no pneumonia or requirement for oxygenation were defined as mildly symptomatic, patients with pneumonia or needing oxygenation were defined as moderately symptomatic, and patients who needed follow-up in the intensive care unit were defined as severely symptomatic. Patients vaccinated with four doses of any COVID-19 vaccine were considered to have received a ‘full dose’ vaccination, as per the requirement by the Ministry of Health of Turkey. The epidemiological data, demographic characteristics, and vaccination information against COVID-19 were recorded.

### 2.2. Blood Samples

A maximum volume of 5 mL of blood was collected from the patients as follows. The admission blood samples (ABSs) were collected immediately after PCR confirmation of infection, which fell within the first two weeks of disease onset, and the control or ‘convalescent’ blood samples (CBSs) were collected at least one week after the ABSs and stored at −80 °C. The variable gap in the collection of the ABSs was related to the (self-)referral of patients prior to PCR diagnosis. The collection of the CBSs also showed a temporal variation related to the patients. However, these were not thought to compromise the integrity of the parameters studied, as the antibody responders of COVID-19 patients were reported to seroconvert 15–30 days post-infection [21]. Patients under the age of 18 and those with a positive COVID-19 PCR test for more than two weeks were not included in the study. Patients who passed away before their CBS could be collected were also excluded from the study.

### 2.3. Serologic Assay

#### 2.3.1. IgM and IgG Measurements

IgG and IgM anti-SARS-CoV-2 RBD antibodies in the ABSs and CBSs were qualitatively measured by lateral diffusion using the COVID-19 IgG/IgM Rapid Test kit (Meril^®^, Cat No: NCVRPD-02, Mumbai, India), following the manufacturer’s instructions. The sensitivity and specificity of the test were reported to be 97.20% and 99.22%, respectively.

#### 2.3.2. Neutralization Antibody Measurements

The Elecsys (Roche^®^) Anti-SARS-CoV-2 S Electrochemiluminescence Immunoassay Kit (Cat NO: 09289267119, Mannheim, Germany) was used for the quantitative in vitro determination of antibodies against the RBD of the SARS-CoV-2 S protein. The assay employs a recombinant form of the RBD of the S antigen to detect all classes of antibodies in a double-antigen sandwich method. Results are reported in units/mL and defined as total neutralizing antibody levels (tNAbs). The specificity and sensitivity of the test were stated to be 100% and 97.92%, respectively. The kit was used in accordance with the manufacturer’s instructions.

In the second part of the study, the blood samples of patients were used for a fusion inhibition test. In this test, the Elabscience (Genscript^®^, Piscataway, NJ, USA) SARS-CoV-2 Surrogate Virus Neutralization Kit was used, and the results were obtained qualitatively. This test is based on competitive ELISA in which tNAbs compete for binding to solid-phase immobilized ACE2 with RBD–horseradish peroxidase conjugate. The assumption is that, among the tNAbs that bind to RBD, only those binding to the ACE2 receptor-binding motif (pNAbs) will yield a positive result. The manufacturer stated that this method has a specificity of 99.93% and a sensitivity of 95–100%.

### 2.4. Statistical Analysis

Statistical analyses were performed using the IBM SPSS Statistics for Windows, Version 22.0 (Armonk, NY, USA: IBM Corp.; 2013). The normality of variables was examined using the Kolmogorov–Smirnov/Shapiro–Wilk tests. Median values, Mann–Whitney U, Wilcoxon and Kruskal–Wallis tests were used for variables not conforming to the normal distribution. Chi-squared and McNemar’s tests were used to compare the differences between the groups.

Correlation analyses were applied to determine the strength and direction of the linear relationship between two variables. Since the data were not normally distributed, Spearman’s rank correlation was used to evaluate the ordinal relationship.

Multinomial logistic regression analysis was performed for multi-category dependent variables. In this analysis, the effect of independent variables on each category was evaluated and the odds ratios were calculated on a category basis. Logistic regression was used for binary dependent variables. In this model, the effect of independent variables on mortality was evaluated and the significance of independent variables was interpreted using the Wald test and odds ratios (Exp(B)). A *p* value of ≤0.05 was considered statistically significant for all types of analysis.

## 3. Results

### 3.1. Demographic Characteristics

In total, the tNAbs of 68 and the pNAbs of 52 patients were analyzed. Of the patients, 35 (51.5%) were female (median age: 67; range: 25 to 94 years). Also, 14 (20.6%) patients were asymptomatic or had a mild infection, 28 (41.2%) had a moderate infection, and 26 (38.2%) had a severe infection. Notably, 41 (60.3%) patients had at least one comorbidity, with cardiovascular disease being the most common (n = 27, 39.7%), followed by essential hypertension (n = 25, 36.8%). The median hospital stay was 17 days (range 1 to 112 days), 21 (30.9%) patients died, and the 28-day mortality rate was 16.2% (n = 11).

### 3.2. IgM and IgG Results

COVID-19 anti-RBD IgM was detected in 22 (32.4%) of the ABSs and 64.7% (n = 44) of the CBSs. Similarly, COVID-19 anti-RBD IgG was detected in 37 (54.4%) of the ABSs and 94.1% (n = 64) of the CBSs.

### 3.3. Neutralization Antibody Results

The tNAb levels in the CBSs (median: 18,140 U/mL; range: 0.4 to 927,920) were found to be statistically significantly higher than those in the ABSs (median: 488.2 U/mL; range: 0.4 to 571,870) (*p* < 0.001). pNAbs were present in the ABSs of 41 (78.8%) patients and in the CBSs of 47 (90.4%) patients; no statistically significant difference was detected (*p* = 0.146).

On the other hand, the tNAb levels in the ABSs were found to be statistically significantly higher in the COVID-19-vaccinated patients compared to those who were not, and in the patients that were positive for IgM or IgG compared to those who were negative for both (*p* < 0.001, *p* < 0.001, *p* < 0.001, respectively). This implies that, if present, IgA anti-RBD had no demonstrable contribution to RBD binding (neutralization). The tNAb levels in the CBSs were statistically significantly higher in patients with hypertension than those without, and in the COVID-19-vaccinated patients than those without (*p* = 0.022 and *p* < 0.001, respectively). No significant relationship was found between the comorbidities or the other parameters investigated and the tNAb levels (*p* > 0.05, Table 1 and Figure 1).

The 28-day mortality rate was statistically significantly lower in patients with pNAbs in both the ABS and CBS compared to those without (*p* = 0.025 and *p* = 0.043, respectively). No significant relationship was found between the other parameters compared and the pNAb levels (*p* > 0.05, Table 2).

### 3.4. Vaccination Results

A total of 49 (72%) patients were vaccinated with any COVID-19 vaccine regardless of type, and 10 (14.7%) of these received the full (four) doses. The vaccination rate with the inactivated vaccine (CoronaVac, Sinovac^®^, Beijing, China) among all the patients was 64.7% (n = 44), while that of the mRNA vaccine (BNT162b2, BioNTech^®^, Mainz, Germany) was 22.1% (n = 15). The median time between SARS-CoV-2 PCR positivity and the last vaccination date of the 49 vaccinated patients was 155 days (range: 3 to 362). Of the vaccinated patients, 20 (40.8%) had a severe infection, 19 (38.8) had a moderate infection, and 10 (20.4%) had an asymptomatic–mild infection, and the 28-day mortality rate was 22.4% (n = 11).

### 3.5. Correlation Analysis Results

Spearman’s correlation analysis revealed significant relationships among the various variables. A negative correlation was found between the presence of pNAbs in the ABS and CBS and the 28-day mortality (r = −0.345, *p* = 0.012 and r = −0.337, *p* = 0.014, respectively). Negative correlations were also found between the full-dose vaccine status and the 28-day mortality (r = −0.331, *p* = 0.002). No correlation was found between the pNAbs and the tNAbs in the CBS (r = 0.193, *p* = 0.169) or the ABS (r = 0.202, *p* = 0.150). Further, there was no correlation between the pNAbs in the ABS and the CBS (r = 0.151, *p* = 0.287). However, there was a positive correlation between the tNAbs in the ABS and the CBS (r = 0.763, *p* = 0.000).

### 3.6. Regression Analysis Results

According to the results of the multinomial logistic regression analysis used to evaluate the independent risk factors thought to be associated with the severity of the disease in hospitalized COVID-19 patients, the probability of the disease being severely symptomatic was 538 times higher than the probability of being asymptomatic or mildly symptomatic in IgG-positive patients on admission (OR 538.7; CI 1.175–247,056.7; *p* = 0.044). In patients with pNAbs on admission, the probability of the disease being asymptomatic or mildly symptomatic was 0.872 times higher (1.14 times less) than the probability of being severely symptomatic (OR 0.872; CI 0.772–0.984; *p* = 0.026, Table 3).

According to the results of the logistic regression analysis used to evaluate the independent risk factors thought to be associated with the 28-day mortality in patients, the presence of pNAbs on the day of admission was shown to be associated with a lower mortality by 0.914 times (OR 0.914; *p* = 0.041). No statistically significant relationship was found between the other parameters thought to add significance to the model and the 28-day mortality (Table 4).

## 4. Discussion

In this study, patients with pNAbs in their ABS were 0.872 times more likely to be asymptomatic or mildly symptomatic (or 1.14 times less likely to be severely symptomatic) compared to being severely symptomatic, and their 28-day mortality was lower than those without pNAbs. The tNAb values were found to be high in both the ABS and CBS from vaccinated patients.

No relationship was found between the tNAb measurements and clinical severity. However, the multinomial regression analysis revealed that in patients with pNAbs in their ABS, the probability of being asymptomatic or mildly symptomatic was greater than the probability of being severely symptomatic. A similar relationship was not observed for pNAbs in the CBS. A review by Mink S. et al. stated that high antibody levels reduced the risk of infection and disease severity [22]. Similarly, Monroe JM. et al. emphasized in a previous study that tNAbs were protective against SAR-COV-2 infection and symptomatic disease development [23]. In our study, the pNAbs in the ABSs support these findings. However, our results do not support these studies in terms of the tNAbs. Takeshita M. et al. reported no tNAb formation in asymptomatic/mildly ill patients [14]. Cavlek et al. noted age-related variability in tNAb titers [24]. Chen W. et al. found that tNAbs may be undetectable in the blood after recovery in some patients [25], which was also the case in this study. Xu J. et al., in a review, stated that the antibody response in severe COVID-19 patients was higher than in asymptomatic patients, but no difference was observed in the early stages of the disease [26]. Shrivasta S et al. showed that tNAbs merely indicated symptomatic infection but seemed unrelated to protection and recovery [15]. Combining our findings with the diverse literature suggests that disease severity cannot be solely attributed to the presence or the titers of tNAbs and that the other contributing factors must be considered (e.g., inadequate innate immune response). These variable results indicate that the clinical condition of the patient will determine the tNAb response [27]. The more severe the disease, the higher the tNAb titers, with rare exceptions. In very few exceptional cases, patients recover or pass without any detectable antiviral antibodies, of which there were two in this study. This hypothesis is further supported by the fact that the neutralizing efficacy of tNAbs is not the sole determining function for antiviral effect [6,8].

When evaluated in terms of mortality, our study showed that the 28-day mortality was lower in patients with pNAbs in their ABS, while there was no significant difference in the tNAb measurements. Numerous studies have shown that higher antibody titers are associated with lower mortality rates [22,28,29]. Our findings regarding pNAbs are consistent with these studies. The pNAbs are expected to limit infection and neutralize all variants of the virus, as well as the wild type. This might ease the disease burden of severely ill individuals (with comorbidities).

As expected, the tNAb titers in the CBSs were statistically significantly higher than those of the ABSs. However, no relationship was found between the tNAbs of either the ABSs or CBSs and the presence of pNAbs. There was no correlation between the tNAbs and the pNAbs. The tNAb and pNAb results in our study were not interrelated, implying that not all tNAbs in our study had pNAb activity.

Another important parameter examined in our study was the relationship between the tNAbs and vaccination. Previous publications have shown that the tNAb levels peak approximately 14 days after vaccination, but only 50% of them can be detected in the blood [8]. Additionally, it has been reported that the tNAb levels in some patients decline to undetectable levels in the blood within six months after the second dose of vaccination [30]. In our patients, no relationship was found between the time of the last vaccination and the tNAbs titers in the vaccinated patients. Studies in the literature summarize that antibody levels against SARS-CoV-2 are low in the naive and unvaccinated groups, but these rates reach 100% after the second dose [8]. In our study, all vaccinated patients received at least two doses of vaccine at least 14 days prior to sampling, except for one individual. In addition, it was observed that tNAbs in the ABS and CBS of vaccinated patients were significantly higher compared to unvaccinated patients. This is in line with the literature. On the other hand, no relationship was shown between the presence of pNAbs and vaccination status in our study. These contradictory findings may be due to the relatively low sample size.

It is known that when patients with hypertension have COVID-19, the disease progresses more severely [31]. Also, studies have shown that COVID-19 significantly affects the cardiovascular system, and one of its most common complications is the emergence of HT [32]. In our patients, the tNAb titers in the CBSs were significantly higher in the HT patients than in those without HT. The reason for this might be that the patients with HT have renin–angiotensin system abnormalities and are more prone to severe infection, and the antibody responses in these individuals may be variable [33].

In this study, the number of participants in the different clinical categories of COVID-19 turned out to be relatively small. One of the reasons for this is that the study was carried out during the mid-to-late stages of the pandemic, and the other is the failure of some of the recruited patients to come to the hospital for the control/convalescent blood collection. This limited or prevented the obtaining of statistically significant results for some parameters and prevented the generalization of the findings. Although the vast majority of patients were vaccinated, the great variability in vaccine types and doses prevented some sub-analyses.

## 5. Conclusions

In this study, as in many others, it is evident that anti-SARS-CoV-2 neutralizing antibody generation is correlated with disease severity and mortality, although this was not consistent in all the cases. It is also supported that all clinical outcomes cannot be attributed solely to the presence or levels of total neutralizing antibodies but partly to pNAbs. Overall, the susceptibility to infection, severe viral illness, and long COVID must be determined primarily by the functionality of the innate immune system, which involves polymorphic MHC antigen presentation and stimulation of cell-mediated immunity.

## Figures and Tables

**Figure 1 life-15-00429-f001:**
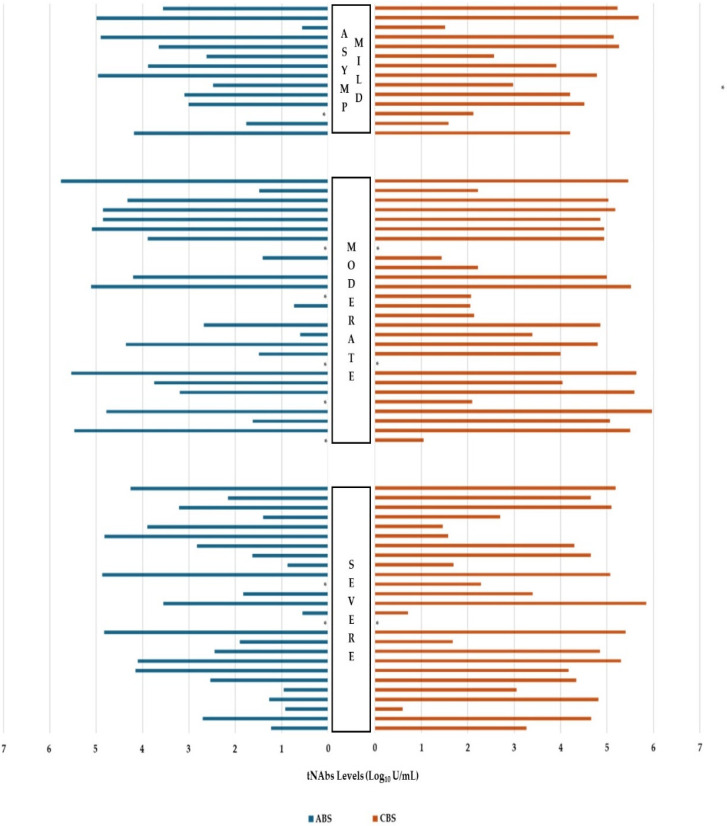
The distribution of admission (ABS) and convalescent/control (CBS) tNAb levels based on disease severity. * Patients with tNAb values of ≤0.4 (negative log), ASYMP: asymptomatic; tNAb: total neutralizing antibody.

**Table 1 life-15-00429-t001:** Comparison of Total Neutralizing Antibody Levels with Epidemiological Data of the Patients.

Parameters				ABS * (n = 68) Median (Min–Max)	*p* Value	CBS * (n = 68) Median (Min–Max)	*p* Value
Gender	Female (n = 35)			477.7 (0.4–343,370)	0.854	11,080 (0.4–927,920)	0.902
	Male (n = 33)			498.7 (0.4–571,870)		32,050 (0.4–474,800)	
Disease severity	Asymptomatic—Mild			2441 (0.4–98,660)	0.496	16,260 (32.49–474,800)	0.699
	Moderate			1027.35 (0.4–571,870)		67,360 (0.4–927,920)	
	Severe			211.75 (0.4–73,640)		17,330 (0.51–706,590)	
COVID-19 Vaccination			No	5.33 (0.4–1619)	<0.001	126.8 (0.4–126,360)	<0.001
			Yes	5610 (0.4–571,870)		66,480 (0.4–927,920)	
Full-dose COVID-19 vaccination (n = 49) ^1^			No	4451 (0.4–571,870)	0.585	72,440 (0.4–927,920)	0.157
			Yes	7758.5 (16.8–90,260)		24,170 (29–170,410)	
Time since LV ^2^ (n = 49)			<6 months	6682.5 (0.4–343,370)	0.758	71,985 (0.4–927,920)	0.743
			>6 months	1577 (4–571,870)		46,110 (37.7–393,440)	
COVID-19 IgM ABS ^3^			Negative	211.75 (0.4–66,090)	<0.001		
			Positive	69,600 (5.33–571,870)			
COVID-19 IgM CBS ^4^			Negative			15,450 (5.07–313,170)	0.488
			Positive			38,145 (0.4–927,920)	
COVID-19 IgG ABS ^3^			Negative	58.1 (0.4–66,090)	<0.001		
			Positive	14,210 (0.4–571,870			
COVID-19 IgG CBS ^4^			Negative			208.09 (5.07–16,290)	0.090
			Positive			27,070 (0.4–927,920)	
28 day mortality		No		677.9 (0.4–571,870)	0.874	16,230 (0.4–927,920)	0.659
		Yes		278.8 (16.8–73,640)		44,240 (29-706,590)	
pNAbs ABS ^3^		No		42.9 (17–343,370)	0.148		
		Yes		5610 (4–571,870)			
pNAbs CBS ^4^		No				117,230 (22,090–706,590)	0.167
		Yes				62,600 (4–927,920)	

^1^ Only vaccinated people were examined, ^2^ Last vaccination, ^3^ Admission blood samples, ^4^ Control blood samples, * ABSs: admission blood samples, CBSs: convalescent blood samples, pNAbs: potential neutralization antibodies.

**Table 2 life-15-00429-t002:** Comparison of epidemiological and laboratory data with patients’ potent neutralizing antibody status.

Parameters			Admission (n = 52)		*p*	Control (n = 52)		*p*
			No	Yes		No	Yes	
Gender (n [%])	Female		8 (30.8)	18 (69.2)	0.174	3 (11.5)	23 (88.5)	1.000
	Male		3 (11.5)	23 (88.5)		2 (7.7)	24 (92.3)	
Age (Year) (median [min-max])			67 (62–85)	67 (28–89)	0.419	77 (65–89)	67 (28–88)	0.232
Hospital stay (median [min-max])			21 (8–83)	15 (1–85)	0.158	21 (14–23)	18 (1–85)	0.664
Disease severity (n [%])	Asymptomatic—Mild		0 (0)	11 (100)	0.089	1 (9.1)	10 (90.9)	0.600
	Moderate		4 (20)	16 (80)		1 (5)	19 (95)	
	Severe		7 (33.3)	14 (66.7)		3 (14.3)	18 (85.7)	
COVID-19 Vaccination (n [%])		No	1 (14.3)	6 (85.7)	1.000	0 (0)	7 (100)	1.000
		Yes	10 (22.2)	35 (77.8)		5 (11.1)	40 (88.9)	
Full-dose COVID-19 vaccination ^1^ (n = 45) [%])		No	8 (22.9)	27 (77.1)	1.000	4 (11.4)	31 (88.6)	1.000
		Yes	2 (20)	8 (80)		1 (10)	9 (90)	
Time since LV ^2^ (n = 45)		<6 months	7 (25.9)	20 (74.1)	0.716	4 (14.8)	23 (85.2)	0.634
		>6 months	3 (16.7)	15 (83.3)		1 (5.6)	17 (94.4)	
COVID-19 IgM ABS ^3^ (n [%])		Negative	8 (25.8)	23 (74.2)	0.491			
		Positive	3 (14.3)	18 (85.7)				
COVID-19 IgM CBS ^4^ (n [%])		Negative				0 (0)	18 (100)	0.150
		Positive				5 (14.7)	29 (85.3)	
COVID-19 IgG ABS ^3^ (n [%])		Negative	7 (36.8)	12 (63.2)	0.074			
		Positive	4 (12.1)	29 (87.9)				
COVID-19 IgG CBS ^4^ (n [%])						0 (0)	2 (100)	1.000
						5 (10)	45 (90)	
28 day mortality (n [%])		No	6 (14.3)	36 (85.7)	0.025	2 (4.8)	40 (95.2)	0.043
		Yes	5 (50)	5 (50)		3 (30)	7 (70)	

^1^ Only vaccinated people were examined, ^2^ Last vaccination, ^3^ Admission blood samples, ^4^ Control blood samples.

**Table 3 life-15-00429-t003:** Results of multinomial logistic regression analysis in evaluating independent risk factors affecting the severity of COVID-19.

Risk Factors			Multinomial Analyse			
				95% Confidence Interval for Exp(B)		
		B	Odd’s Ratio	Lower Bound	Upper Bound	*p* Value
Moderate symptomatic	IgM admission	−3.521	0.03	0.001	1.352	0.071
	IgG admission	5.213	183.6	0.546	61,699.735	0.079
	Days since LV ^1^	0.006	1.006	0.992	1.019	0.419
	tNAbs admission	0.000	1.000	1.000	1.000	0.184
	tNAbs control	0.000	1.000	1.000	1.000	0.240
	pNAbs admission	−0.098	0.907	0.806	1.021	0.107
	pNAbs control	0.040	1.041	0.983	1.102	0.174
Severe symptomatic	IgM admission	−0.485	0.616	0.018	30.948	0.787
	IgG admission	6.289	538.7	1.175	247,056.786	0.044
	Days since LV ^1^	0.007	1.007	0.993	1.021	0.350
	tNAbs admission	0.000	1.000	1.000	1.000	0.708
	tNAbs control	0.000	1.000	1.000	1.000	0.310
	pNAbs admission	−0.137	0.872	0.772	0.984	0.026
	pNAbs control	0.021	1.021	0.974	1.071	0.388

^1^ last vaccination.

**Table 4 life-15-00429-t004:** Results of logistic regression analysis in evaluating independent risk factors affecting 28-day mortality in patients.

Risk Factors	Logistic Analysis		
	B	Odd’s Ratio	*p* Value
IgM admission	−2.501	0.082	0.402
IgM control	−0.175	0.840	0.915
IgG admission	−3.491	0.03	0.341
IgG control	−15.590	0.000	1.000
Days since LV ^1^	−0.013	0.987	0.161
tNAbs admission	0.000	1.000	0.141
tNAbs control	0.000	1.000	0.301
pNAbs admission	−0.089	0.914	0.041
pNAbs control	−0.004	0.006	0.792

^1^ last vaccination, tNAbs: total neutralizing antibodies, pNAbs: potential neutralising antibodies.

## Data Availability

Original data supporting the findings of this study are available.

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
