# Peer review of "Clinical Significance of Neutralizing Antibodies in COVID-19: Implications for Disease Prognosis"

_life, 2025, doi:10.3390/life15030429_

Round 1
Reviewer 1 Report
Comments and Suggestions for Authors
This is a well designed and reported study. I have no major concerns.
Minor concerns: very minor grammar errors, I think there is a missing word at line 59 (that?)
Comments on the Quality of English LanguageAs above, well written but some minor grammatical errors
Reviewer 2 Report
Comments and Suggestions for Authors
The manuscript life-3464431 entitled Clinical Significance of Neutralizing Antibodies in COVID-19: Implications for Disease Prognosis by Sudem MahmutoÄŸlu Çolak and collaborators aims to investigate the relationship between total NAb (tNAb) levels, presence of potential neutralization inhibiting antibodies (pNAbs), and the clinical outcomes of COVID-19 patients.
Patients aged ≥18 years diagnosed between October 2021 and September 2022 were grouped by symptom severity. Two time points blood sampling and data on demographics, epidemiology, and vaccination were recorded.
The tNAbs and pNAbs were measured by enhanced chemiluminescence assay and surrogate virus neutralization test, respectively.
Among 68 tNAbs and 52 pNAbs results, 26 (38.2%) patients had a severe infection. The 28-day mortality rate was 16.2% (n=11).
The tNAbs levels in the control blood samples (CBS) were significantly higher than those of the admission blood samples (ABS) (p<0.001). The statistical analysis showed no relation between disease severity and tNAb titers. Vaccinated patients had significantly higher tNAbs in ABS and CBS (p<0.001 and p<0.001, respectively). Presence of pNAbs in the ABS correlated with lower 28-day mortality (p=0.026) and milder disease course (p=0.041).
The work is scientifically well constructed and investigated the connection between total and neutralizing antiboides in Sars-CoV2 infection.
The experimental work was properly planned and conducted
Results are clear and properly statistically studied.
Tables are informative.
Discussion is consistent with results.
References are appropriated.
A minor linguistic revision is recommended.
Line 91: serologic
Line 101 and line 110: more information about precision, reproducibility, between run variation should be added for the kits used.
Line 159: the table 1 should be in one page.
Line 159: table 1: add the number of males and females
Line 167: table 2 should stay in one page.
Line 167: table 2: add the number of males and females
Line 197: table 3 in one page
Reviewer 3 Report
Comments and Suggestions for Authors
The study addresses an interesting and relevant topic; however, significant improvements are needed to meet optimal scientific standards:
1. Terms related to COVID-19 should be used consistently throughout the manuscript (e.g., “COVID-19” instead of “Covid-19”).
2. Tables 1 and 2 are difficult to follow and require a clearer or more simplified presentation.
3. The article lacks graphical elements, which would enhance data interpretation.
4. Not all obtained results are discussed in relation to existing medical literature, limiting the relevance of the conclusions.
5. The number of patients included in the study is too small to support robust conclusions, necessitating a larger sample for validation.
Comments on the Quality of English LanguageSome sentences are awkwardly constructed and require reformulation for clarity. There are grammatical errors that need immediate attention. The scientific tone is not consistently maintained.
Reviewer 4 Report
Comments and Suggestions for AuthorsÇolak et al studied the effect of neutralizing antibodies in COVID-19 disease severity and outcome. They showed that total NAB (tNab) titre has no correlation with disease severity.
However, the paper is difficult to follow, mainly due to poor English usage that hindered the significance of the study. The paper seems to me a draft and contains numerous conceptual mistakes and should be written freshly.
There are some examples.
1. In abstract, Line 22-23, authors stated that there is no relation between tNabs and disease severity. But in Line 26-27, “these finding support a correlation between tNabs and disease severity and mortality”!
2. In introduction, authors should explain “potential neutralization inhibiting antibody” (pNabs) and how does it differ from total antibodies (tNabs)
3. Line 27, what is formation?
4. Line 40, what are INF I &III? If it is interferon they are generally written as IFN not INF.
5. Line 49, NAbs (here referred to native antibodies) are raised by the viral infection. Then how do the same virus can evade it by mutation? Authors should provide references.
6. Line 54, what does it mean by “Nab levels are elevated in individuals exhibiting severe disease symptoms”? line 55 does not make any sense.
7. Line 65-66, “The vast majority of people who fail to become ill post-viral exposure and that those who become mildly ill lack NAbs”-how does authors get this information which is a totally wrong.
8. In line 83, authors stated that they considered only people with four doses of vaccines as full doses of vaccine. Which vaccine needs four doses to be complete vaccination? Again, in line 172, wrote forty-nine (72%) patients received two doses of vaccine!
9. Division of ABS (Admission Blood Sample) and CBS is problematic. ABS is within two weeks of admission with RT-PCR positive. But in what logic, CBS (control Blood Sample) after one week of collection of ABS was done?
10. And so on.

The usage of English sentences constructions is poor. The paper also seems to be a draft. Extensive editing is required.
Reviewer 5 Report
Comments and Suggestions for Authors
Colak et al present a well-structured and thought-out study investigating the relationship between total neutralizing antibody (tNAb) levels, potential neutralization-inhibiting antibodies (pNAbs), and the clinical outcomes of COVID-19 patients. Overall the work has an interesting conclusion: that a key set of antibodies are more important than the total set. This fills an interesting niche in the COVID-19 literature. Overall I have only minor suggestions to improve the manuscript, which is well written but does not discuss in much detail how the work relates to the COVID-19 immunology field.
1. Abstract
o I’m not quite sure what: "These findings support a correlation between tNAbs and disease severity and mortality, but this seems related to their formation rather than their effect on clinical outcomes" could be revised for better readability.
2. Introduction
o The introduction needs a brief summary of the findings of the paper.
o “Moreover, the vaccine effectiveness is better against symptomatic infection than asymptomatic infection” – I don’t fully understand how this would be measured.
3. Methodological issues
o Was there any exclusion criteria for the patients?
o I would state how much blood was taken from each patient for the study.
o It is slightly odd to consider “full dose” as four doses of vaccine rather than, say, two. Is there a reason for this?
o The study mentions that vaccinated patients had significantly higher tNAbs but does not control for the time elapsed since the last vaccination. Given the waning nature of vaccine-induced immunity, this is a crucial factor that should be accounted for in the statistical analysis.
4. Results/Statistics:
o The correlation between pNAbs at admission and 28-day mortality is an interesting finding. The discussion should consider potential mechanisms for why pNAbs might be more predictive of survival than tNAbs.
o Table 1 needs a clear heading to state that it is measuring antibody levels. It is hard to interpret otherwise.
5. Discussion:
o The authors conclude that the innate immune system also plays a role, but do not discuss the role of T cells in controlling SARS-CoV-2. A review like this would be helpful for the authors to read: 10.1126/sciimmunol.abo1303.
o This manuscript only discusses acute disease but immunity also plays a role in persistent symptoms. Some suggest quite large changes: Lancet Infect Dis 2022;22:43–55. Clin Infect Dis 2023;76:738–40. Open Forum Infectious Diseases 2022;9:ofac464. Kuodi, npj Vaccines 7, 101 (2022); others suggest smaller changes: 10.1038/s41591-022-01840-0. The authors ought to mention this.
Author Response
Please see the attached cover letter.

Round 2
Reviewer 3 Report
Comments and Suggestions for Authors
Significant modifications have been made to the study, which marks an important step towards its publication. However, to enhance its scientific rigor and relevance, the following adjustments are necessary:
1. Inclusion of bibliographic references at lines 107, 115, and 123 to support the statements made in the text.
2. Enhancement of the statistical analysis section by providing the formula used to calculate the sample size, thereby strengthening the scientific validity of the results.
3. Mention of the study's practical implications in the conclusions and discussion sections, emphasizing its clinical impact and applicability.
Reviewer 4 Report
Comments and Suggestions for Authors
Authors provided explanations for the comments raised earlier. Authors also argued (answers to comment 9) that papers with similar content exist in the literature. But what authors did not understand that it is writing of this paper which is not as the standard of scientific reporting. Authors must consider professional writer or native English writer to edit the manuscript.
For example,
- In response to comment 2 and that is incorporated in text (line 59-63). In response, the first two sentences are correct as a definition of tNAB. But when they wrote the statement in comments a well as in the text, the meaning is reversed. It meant that tNAB binds RBD but some portion fails to hinder…whereas pNAB potentially binds all! Based on this reverse meaning, whole results are reversed!
- Response to comment 5, these summary of the explanations ( as they wrote in Line 54 (Srivastava et al and Garcia-Barton et al, Line 55, Franchini and Focosi) should be incorporated in the manuscript with the references they provided.
- In response to comments 7, authors should write in the manuscript that 4 doses of vaccines were considered as full vaccination as per their country’s Ministry of health rule. Because it is unusual.
- Answers to comments 8 is not satisfactory. However, authors should provide these explanations in the text of the manuscript why they used CBS after one week of ABS.
Comments on the Quality of English Language
Authors should rewrite some of the sentences and extensively edit the manuscript to understand the meaning is correct what they want to say.
